# `Text2Tree`: Aligning Text Representation to the Label Tree Hierarchy for Imbalanced Medical Classification

**Jiahuan Yan**[1], **Haojun Gao**[2], **Kai Zhang**[2], **Weize Liu**[3], **Danny Chen**[4], **Jian Wu**[2,*] **Jintai Chen**[1,5,*]

[1]College of Computer Science and Technology, Zhejiang University, Hangzhou, China

[2]School of Medicine, Zhejiang University, Hangzhou, China

[3]Polytechnic Institute, Zhejiang University, Hangzhou, China

[4]Department of Computer Science and Engineering, University of Notre Dame, IN, USA

[5]Computer Science Department, University of Illinois Urbana-Champaign, IL, USA

{jyansir, joagh, zhangkai1999, weizeliu, wujian2000}@zju.edu.cn, dchen@nd.edu, jtchen721@gmail.com

## Abstract

Deep learning approaches exhibit promising performances on various text tasks. However, they are still struggling on medical text classification since samples are often extremely imbalanced and scarce. Different from existing mainstream approaches that focus on supplementary semantics with external medical information, this paper aims to rethink the data challenges in medical texts and present a novel framework-agnostic algorithm called `Text2Tree` that only utilizes internal label hierarchy in training deep learning models. We embed the ICD code tree structure of labels into cascade attention modules for learning hierarchy-aware label representations. Two new learning schemes, **S**imilarity **S**urrogate **L**earning (SSL) and **D**issimilarity **M**ixup **L**earning (DML), are devised to boost text classification by reusing and distinguishing samples of other labels following the label representation hierarchy, respectively. Experiments on authoritative public datasets and real-world medical records show that our approach stably achieves superior performances over classical and advanced imbalanced classification methods. Our code is available at https://github.com/jyansir/Text2Tree.

## 1 Introduction

Medical text classification is widely recognized as an urgent yet challenging problem due to its extremely imbalanced data distribution, large variety of rare labels (Johnson et al., 2016; Ziletti et al., 2022), and complicated label relationship (Tsai et al., 2021; Vu et al., 2021). Various downstream clinical tasks have been derived from this problem, including ICD coding (Mullenbach et al., 2018; Yuan et al., 2022; Yang et al., 2022) and automated diagnosis (Chen et al., 2020b), showcasing its potential values in modern clinical practice with machine learning approaches (Berner, 2007).

There have been various classical strategies for general imbalanced classification and long-tailed multi-label classification (Huang et al., 2021) in machine learning studies. Such studies probably overly focused on data in rare categories (e.g., by re-sampling (Chawla et al., 2002; Menardi et al., 2014), re-weighting (Kumar et al., 2010; Lin et al., 2017; Chang et al., 2017; Li et al., 2020; Wang et al., 2022a), ensemble learning (Khoshgoftaar et al., 2007; Liu et al., 2020), data augmentation (Mariani et al., 2018; Chu et al., 2020; Zada et al., 2022)) to alleviate distribution bias, or employed sophisticated learning paradigms such as contrastive learning (Wang et al., 2021), transfer learning (Ke et al., 2022), and prompt-tuning (Zhang et al., 2022)) to learn fine-grained representations of hard cases. But, in medical text classification, many rare diseases lack sufficient data in representation learning and the incidence rates vary greatly for different diseases, which cannot be completely solved by the general imbalanced classification approaches since they overlook the underlying dependency of medical terminologies.

Although the current studies of medical text classification either leveraged the label descriptions (Chen and Ren, 2019; Zhou et al., 2021; Yang et al., 2022) or label dependency (Xie et al., 2019; Cao et al., 2020) for precise pairwise sample-label matching (Mullenbach et al., 2018), or incorporated external knowledge sources (e.g., Wikipedia) to enrich semantic information (Prakash et al., 2017; Bai and Vucetic, 2019; Wang et al., 2022b), they did not explicitly cope with the data imbalance and scarcity issues. Therefore, imbalanced medical text classification is still an open challenge.

Targeting this problem, this paper makes the first effort on learning medical text representations to resolve the data imbalance and scarcity issues with the support of disease label dependency. As shown in Fig. 1, the right part demonstrates an example of ICD-10-CM codes (a clinical modi-

---

*Corresponding authors.

fication of ICD-10 codes) about *pneumonia* and *hypertension*-related disease dependency in a tree structure. Among all the types of *pneumonia*, *fungal pneumonia* stands out as an exceptionally rare and life-threatening variant (Meersseman et al., 2007) that poses significant challenges to early detection (Morrell et al., 2005). Consequently, diagnosed cases of *fungal pneumonia* are notably scarce, and we aim to supplement our analysis with existing samples from the other diseases. An intuitive assumption is that two diseases with some common clinical presentations (e.g., symptom, disease course, treatment) can be used to compensate for each other. Naturally, we can associate *fungal pneumonia* (J16.8) with *COVID-19* (J12.82), since both of them are respiratory system diseases (J00-J99) and pneumonia (J09-J18) while the diagnosed cases of *COVID-19* are plentiful. Similarly, *chlamydial pneumonia* (J16.0) shares a closer parent node (J16) and can be a better supplementary source, while some more dissimilar ones (*e.g.*, circulatory system diseases, I00-I99) are not so beneficial. Utilizing the tree hierarchy, we are able to access prior knowledge of the similarity between disease types, which is helpful in dealing with the data imbalance and scarcity issues.

Motivated by this, we present a new framework-agnostic algorithm that aligns **text** representation **to** label **tree** hierarchy (called `Text2Tree`), which is tailored for better classification by incorporating an ICD code embedding tree to guide medical text representations. `Text2Tree` has three major components: (i) a *hierarchy-aware* label representation learning module (HLR), (ii) *similarity* surrogate learning (SSL), and (iii) *dissimilarity* mixup learning (DML) approaches. Unlike general imbalanced classification methods that compensate a sample by reusing samples with the same disease labels or itself, (ii) and (iii) are designed to reuse samples from other labels under the specification of the label hierarchy learned by (i). In this paper, the used "codes" refer to disease labels with underlying hierarchy, and we do not distinguish "code", "label", and "node".

The `Text2Tree` training procedure proceeds as follows. First, cascade attention modules are built based on the prior structure of the medical ICD code tree. Only embeddings of labels on adjacent tree layers are interacted in each module, and label representations are computed layer by layer. Hence, a strict interaction fashion is kept to constrain in-

formation flow along tree edges. Next, pairwise label similarity is calculated based on the label representations. Based on the similarity, samples from other labels will be treated as surrogate positive anchors to provide extra contrastive signal by SSL, or apply mixup to give new samples by DML. The SSL branch learns to gather more information from more similar labels to explicitly reuse texts in representation learning following the label hierarchy, while DML generates and classifies hard cases, adversarially preventing excessive manifold distortion resulted from SSL.

Our main contributions are as follows:

- We first explore the medical text representation learning problem from the data imbalance and scarcity perspective, and propose the new `Text2Tree` algorithm that aligns text representations to the label tree hierarchy.
- In contrast to previous methods that reused samples with the same labels, we propose SSL and DML to leverage samples from diverse labels to facilitate representation learning based on the underlying label hierarchy.
- Comprehensive experiments show that our `Text2Tree` algorithm stably outperforms advanced framework-agnostic imbalanced classification algorithms, without any external medical resource.

## 2 Related Work

### 2.1 General Imbalanced Classification

Imbalanced classification refers to a ubiquitous machine learning problem in practical applications with long-tailed data distribution (Sun et al., 2009; Zhang et al., 2023). A major category of classical algorithms attempts to alleviate this problem from the data perspective, such as re-sampling (Menardi et al., 2014), re-weighting (Lin et al., 2017), ensemble learning (Liu et al., 2020), and data augmentation (Chu et al., 2020; Zada et al., 2022). The core essence of these methods is to fix the skewed distribution or thoroughly harness the available data, and thus they can be easily adopted in both statistical machine learning and deep learning paradigms.

Another line of work seeks to address class imbalance at the data representation level. A common assumption is that better feature representations make better classifiers (Wang et al., 2021), which is intuitive for deep learning models. Among these methods, prototype learning (Liu et al., 2019; Zhu

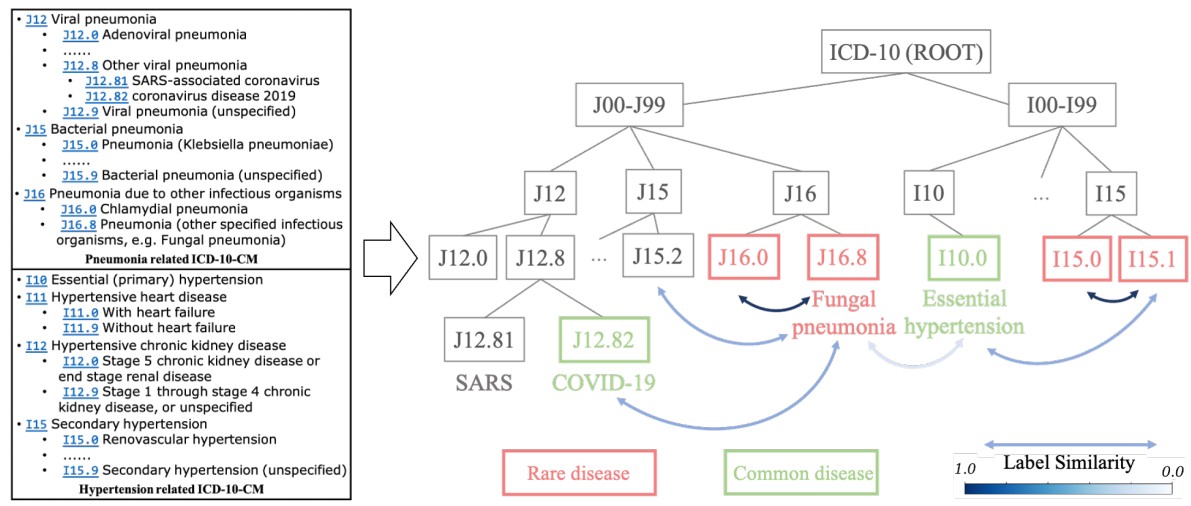

Figure 1: Hierarchical ICD-10 codes on pneumonia related and hypertension related diseases (left) can be organized into a tree structure (right). The code tree contains prior knowledge on disease label similarity that can guide data reuse from other labels. Here "similarity" is a soft metric that can be computed according to the code tree.

and Yang, 2020) and contrastive learning (Wang et al., 2021; Kang et al., 2021) select reasonable anchor samples for feature alignment; transfer learning (Cui et al., 2018; Kang et al., 2021; Ke et al., 2022) acquires a better initial representation based on pre-training or samples from other domains. More recently, prompt-tuning (Zhang et al., 2022) is also a trending technique to take advantage of representation ability of large pre-trained models.

## 2.2 Hierarchical Text Classification

Hierarchical text classification (HTC) is a challenging sub-field of multi-label text classification (Wehrmann et al., 2018; Chen et al., 2021; Wang et al., 2022c,d; Jiang et al., 2022). The labels of HTC fall under different levels in a label tree, while medical text classification is a flat classification problem (each label in ICD coding is a specific disease at the lowest level) though in this paper, an underlying label hierarchy exists based on the medical code tree.

HTC methods can be categorized into local and global approaches. The local ones (Wehrmann et al., 2018; Shimura et al., 2018; Banerjee et al., 2019) leveraged label hierarchical information to separately build a classifier for each label level in the label tree. Currently, the global series become prevalent for their better performance (Chen et al., 2021). Methods of this type treated HTC as a multi-label text classification problem on all the nodes in the label tree, and the main concern is to propose effective frameworks for better hierarchy encoders and label representation. (Zhou

et al., 2020) first introduced prior hierarchy knowledge with structure encoders for modeling label dependency in HTC, (Chen et al., 2021) further performed label-text semantic matching in a joint embedding space to distinguish target labels from incorrect labels, (Wang et al., 2022d) incorporated a contrastive learning framework by masking unimportant tokens to generate positive samples from the original ones, and (Wang et al., 2022c) used soft prompts to fuse label hierarchy into pre-trained models for better adaption to HTC. Recent studies (Jiang et al., 2022) also used combination of local and global views to take advantage of both types of approaches.

## 3 Methodology

In this section, we separately describe the three key components of our proposed Text2Tree algorithm: *hierarchy-aware* label representation (HLR) learning module, *similarity* surrogate learning (SSL), and *dissimilarity* mixup learning (DML).

## 3.1 Hierarchy-aware Label Representation

Given prior label dependency (e.g., an ICD code tree as shown in Fig. 1), it is intuitive to encode hierarchy information with a graph encoder. Different from the recent work on HTC (Wang et al., 2022d) that used complicated neural architectures (e.g., GraphFormer (Ying et al., 2021)), here we propose a cascade tree attention module to introduce dependency bias among labels. Given a code tree of maximum level $L$ (we define the ROOT to be at level 0), we model a representation of

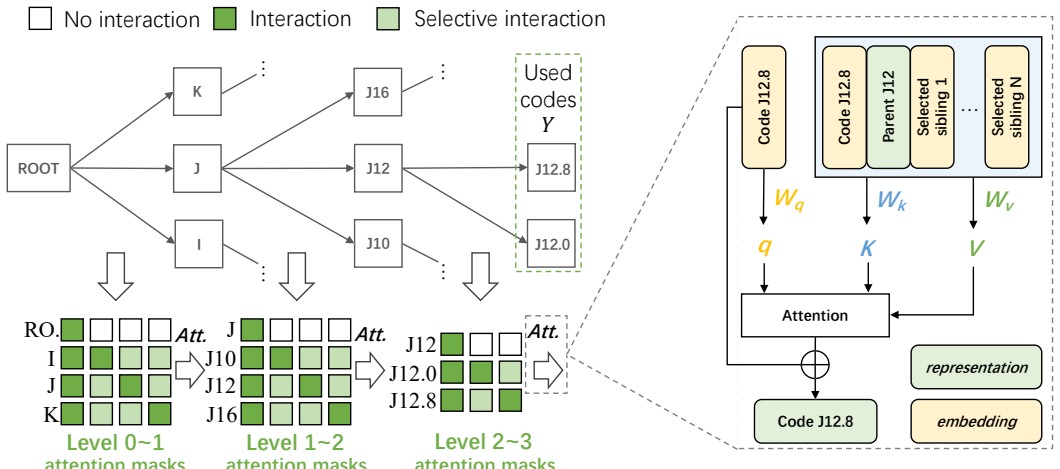

**(a) The computing process of cascade attention layers**

**(b) Computing J12.8 label representation from related labels**

Figure 2: The HLR module consists of cascade attention layers that derive attention masks from the code tree hierarchy (left). The representation of a label contains information from itself, its parent and selected siblings (right). "Used codes $Y$" defines the disease label set (at the lowest level of an ICD tree) for classification.

code $i$ from its parent code $p$ and sibling codes $S \equiv \{j \mid \text{parent}(j) = \text{parent}(i) = p, j \neq i\}$. We formulate the code $i$ representation $h_i$ as follows:

$$q_i = W_q e_i, k_i = W_k e_i, v_i = W_v e_i, \quad (1)$$

$$k_j = W_k e_j, \ v_j = W_v e_j, \epsilon_j = \text{sigmoid}(s_j), \quad (2)$$

$$k_p = W_k h_p, v_p = W_v h_p, \quad (3)$$

$$K_i = [k_i, k_p, K_J], V_i = [v_i, v_p, V_J], \quad (4)$$

$$h_i = \text{Attention}(q_i, K_i, V_i) + e_i. \quad (5)$$

In Eq. (4), $J = S \cap \{j \mid \epsilon_j > 0.5\}$ is the selected sibling subset, $K_J$ and $V_J$ are key and value matrices composed of the corresponding vectors in Eq. (2). Inspired by a recent Transformer architecture with selective feature interactions (Yan et al., 2023), we design a sibling selector $\epsilon_j$ in Eq. (2) by performing activation on a learnable parameter $s_j$, indicating whether the sibling $j$ should interact with node $i$ in the attention module to achieve data-driven sibling aggregation. The core of the whole process is to attentively fetch information from the parent label and selected sibling labels to express the representation of label $i$ (see Fig. 2). Here $e_i$ denotes the learnable embedding of label $i$, and $W_q$, $W_k$, and $W_v$ represent transformations to the query, key, and value vectors in attention mechanism. Note that in Eq. (3), we use the parent node representation $h_p$ to generate its key and value vectors. The reason for this is that we process label representations at different levels layer by layer (see the left part of Fig. 2). The first attention layer only contains level-1 (L1) labels and the ROOT, where the interactions between L1 labels and the

ROOT are compulsory, and the ones between an L1 label and its siblings are selective. Similarly, the second attention layer only includes level-2 (L2) and L1 codes, and when we calculate the representation of an L2 label, using its parent (an L1 label) representation $h_p$ rather than embedding $e_p$ helps pass information from the higher levels (the ROOT and the parent's siblings). There are $L$ cascade attention layers in total. In Eq. (4), the key and value matrices of label $i$ are composed of the corresponding vectors of itself, its parent $p$ (compulsorily included) and siblings in $J$ (selectively included by threshold clipping on sibling selectors $\epsilon_j$). In Eq. (5), the final label representation $h_i$ is composed of attentively fetched information and its own embedding $e_i$.

We define the representation and embedding of the ROOT as equal ($h_{\text{ROOT}} = e_{\text{ROOT}}$) since the ROOT has no parent and siblings. The straight-through trick (Bengio et al., 2013) is used to solve the undifferentiable issue of selecting siblings in Eq. (4). Before training, all label embeddings $e$ can be randomly initialized or assigned with average of BERT token embeddings if label texts are given.

### 3.2 Similarity Surrogate Learning

Contrastive learning has been extensively validated as an effective representation learning method in the image (Chen et al., 2020c; He et al., 2020) and text (Gunel et al., 2021) domains. Prevailing contrastive methods can be roughly categorized into self-supervised and fully-supervised ones (Khosla et al., 2020). The basic idea of both types is to

slightly adjust the distance between samples and anchors for a better data representation space. A classical form of the supervised contrastive learning (SCL) loss is:

$$L_{scl} = \sum_{i \in I} \frac{-1}{|P(i)|} \sum_{p \in P(i)} \log \frac{\exp(z_i \cdot z_p/\tau)}{\sum\limits_{a \in A(i)} \exp(z_i \cdot z_a/\tau)},$$
(6)

where $i \in I \equiv \{1, \ldots, N\}$ denotes the sample index in a batch (or dataset), $A(i) \equiv I \setminus \{i\}$, $P(i) \equiv \{p \mid p \in A(i), y_p = y_i\}$, $z_i$ is the encoded features of sample $i$, and $\tau \in \mathcal{R}^+$ is a scalar temperature. In this paper, we use the hidden state of the first token ([CLS]) after the BERT encoder (i.e., $z_i = \text{BERT}(x_i)_{[CLS]}$). Eq. (6) intuitively extends positive anchors from self-generated samples in an unsupervised manner to samples with the same label in a supervised manner, thus effectively leveraging label information (Khosla et al., 2020). But, rare labels are common in medical text classification. For example, in the top-100 frequent labels of the PubMed dataset, the top-10 labels account for 40.2% of the total label amount, while the rarest 50 labels occupy only 23.8% (see Fig. 5, Appendix A). Among 8,692 unique ICD-9 codes of the MIMIC-III dataset (Johnson et al., 2016), 4,115 codes occur less than 6 times (Yang et al., 2022). In our real-world datasets, there is also a large amount of rare diseases (see the figures in Appendix A). Samples with rare labels are unlikely to match any positive sample in a batch (the value $|P(i)|$ tends to be zero in Eq. (6)), making it **intractable** to acquire contrastive signal for such samples.

To tackle this problem, we propose Similarity Surrogate Learning (SSL) based on the *label similarity*, which treats any other sample in a batch as a potentially positive surrogate anchor. Specifically, we define the similarity score between samples $i$ and $j$ based on their label representations:

$$\text{Sim}(i, j) = \frac{h_{y_i}^T h_{y_j}}{\|h_{y_i}\|_2 \|h_{y_j}\|_2},$$
(7)

where $h_{y_i}$ is the label representation of sample $i$. For multi-label classification, we calculate the average representation $h_{Y_i}$ of all the codes, as:

$$h_{Y_i} = \frac{1}{|Y_i|} \sum_{y \in Y_i} h_y.$$

Here, we simply choose the cosine similarity. We further extend Eq. (6) based on the similarity score:

$$f(i, j) = \text{Sim}(i, j) \cdot \log \frac{\exp(z_i \cdot z_j/\tau)}{\sum\limits_{a \in A(i)} \exp(z_i \cdot z_a/\tau)},$$

$$L_{ssl} = \sum_{i \in I} \frac{-1}{\sum\limits_{a \in A(i)} \text{Sim}(i, a)} \sum_{a \in A(i)} f(i, a), \quad (8)$$

where $f(i, j)$ is a soft contrastive term between sample $i$ and its surrogate anchor $j$, which encourages the algorithm to contrastively reuse samples with high label similarity $\text{Sim}(i, j)$. Label representations $h$ are learnable in Sec. 3.1, and thus the surrogate anchor selection is data-driven.

Actually, our SSL loss (Eq. (8)) can be viewed as a combination of the SCL loss and an extra term:

$$L_{ssl} = \sum_{i \in I} \frac{-1}{\sum\limits_{p \in P(i)} \text{Sim}(i, p)} \sum_{p \in P(i)} f(i, p)$$
$$+ \sum_{i \in I} \frac{-1}{\sum\limits_{b \in B(i)} \text{Sim}(i, b)} \sum_{b \in B(i)} f(i, b)$$
$$\equiv L_{left} + L_{right}, \quad (9)$$

$$\because \forall p \in P(i), y_p = y_i,$$
$$\therefore \text{Sim}(i, p) \equiv 1 \Rightarrow \sum_{p \in P(i)} \text{Sim}(i, p) \equiv |P(i)|,$$
$$\therefore L_{left} \equiv L_{scl} \Rightarrow L_{ssl} \equiv L_{scl} + L_{right},$$

where $B(i) \equiv A(i) \setminus P(i)$ denotes samples with different labels from sample $i$. Obviously, our SSL loss leverages samples with other labels to contribute contrastive signal (since $L_{right} > 0$ (Eq. (9))), and such extra signal is likely to relief the data scarcity issue on rare disease labels by alleviating the sparsity of optimization signal. Overall, Eq. (9) indicates a progressive relationship that SSL is a generalized form of SCL, with extra flexibility of user-defined sample similarity function (Eq. (7)) based on the specific real-world application. SSL is approximately equivalent to SCL for a sample in common labels, while for a sample in the low resource scenario (e.g., rare disease, online learning, small batch size for limited hardwares), SSL makes contrastive signal tractable for such hard cases.

## 3.3 Dissimilarity Mixup Learning

As a typical interpolation-based augmentation technique, Mixup (Zhang et al., 2018) has been widely adapted to NLP settings (Chen et al., 2020a; Sun et al., 2020) and proved to be an effective data-adaptive regularization to reduce overfitting (Zhang

et al., 2021). We exploit this strategy to reuse information of samples with less similar labels by introducing Dissimilarity Mixup Learning (DML). Different from the ordinary Mixup strategy that samples weights from the Beta distribution ($\lambda \sim \text{Beta}(\alpha, \alpha)$), we directly assign the weights according to the similarity score (Eq. (7)), as:

$$\lambda = 0.5(1 + \text{Sim}(i, j)), \qquad (10)$$
$$\tilde{z} = \lambda z_i + (1 - \lambda)z_j,$$
$$\tilde{y} = \lambda y_i + (1 - \lambda)y_j,$$

where $z_i$ is the same as that in Sec. 3.2, and $y_i$ is in the one-hot representation. DML is driven by label representations learned in Sec. 3.1, and tends to mix a bigger portion of more dissimilar samples, so as to generate hard samples. Notably, our DML needs no hyperparameter compared to the ordinary Mixup, and is more user-friendly.

### 3.4 The Overall Training

Prediction is made based on a BERT pooler and a simple fully connected layer after mixing the encoded features. The final loss function is the combination of classification loss and the SSL loss:

$$\hat{y} = \text{FC}(\text{BertPooler}(\tilde{z})),$$
$$L = (1 - \lambda)L_{ce} + \lambda L_{ssl}, \qquad (11)$$

where $L_{ce}$ is cross entropy loss for multi-class classification and is binary cross entropy loss for multi-label classification, and $\lambda$ is a hyperparameter controlling the loss weight. In backward propagation, we detach the gradient in Eq. (10) and only optimize HLR through Eq. (8). We illustrate and discuss backward gradient flow in Appendix D.

## 4 Experiments

### 4.1 Experimental Setup

**Datasets and Evaluation Metrics.** We experiment on two authoritative medical text datasets: MIMIC-III (Johnson et al., 2016) and PubMed[1], and three in-house real-world datasets: Dermatology, Gastroenterology, and Inpatient. Here, the first two public datasets are for multi-label classification, and the three in-house ones are for multi-class classification. In experiments, for MIMIC-III, we use only 33 disease labels in the top-50 version, and convert ICD-9 codes into ICD-10 codes. For

[1]https://www.kaggle.com/datasets/owaiskhan9654/pubmed-multilabel-text-classification

PubMed, we use a recent Kaggle version since it is well sorted and contains 50K research articles from the PubMed repository, and we retain the top-100 3-level MeSH labels (i.e., each level of MeSH ID "C01.784" is "C", "C01" and "C01.784"). Dermatology and Gastroenterology are two datasets cleaned from outpatient records of the two largest departments in a top hospital during the last three years, and Inpatient is cleaned from the inpatient records from a famous healthcare institution. All the three real-world datasets are annotated with ICD-10 codes, and two clinical graduates cleaned them according to the discipline in Appendix B. For data splitting, we use 20% samples as test set, 16% as evaluation set, and the rest 64% as training set. For each multi-class dataset, we further keep the ratios of each label in the three sets the same.

The dataset statistics are given in Table 1. More detailed dataset information is provided in Appendix A.

The metrics Macro-F1 and Micro-F1 are used for measuring the classification results.

| Dataset | $N$ | $|Y|$ | $\text{Avg}(l_i)$ | $\text{Avg}(|y_i|)$ |
|---|---|---|---|---|
| MIMIC-III | 11.4K | 33 | 450.61 | 4.01 |
| PubMed | 50.0K | 100 | 122.88 | 8.52 |
| Dermatology | 20.5K | 59 | 44.97 | - |
| Gastroenterology | 35.0K | 35 | 58.31 | - |
| Inpatient | 2.6K | 98 | 69.22 | - |

Table 1: Dataset statistics. $N$ is the number of samples, $|Y|$ is the number of classes, and $\text{Avg}(l_i)$ and $\text{Avg}(|y_i|)$ are the average token length and average label amount per sample, respectively.

**Baselines.** For systematic and fair comparison, we utilize various framework-agnostic imbalanced classification algorithms and advanced hierarchical text classification (HTC) methods. 1) **Finetune**: The ordinary finetune paradigm. 2) Resampling: A classical method to alleviate distribution bias; we use **RandomOverSample** (ROS) implemented in the *Imbalanced-learn* Python package for multi-class tasks and **distribution balance loss** (DBLoss) designed by (Huang et al., 2021) for multi-label tasks, because the ordinary re-sampling methods are not effective. 3) Re-weighting: We choose **FocalLoss** (Lin et al., 2017) since it is a typical dynamically weighted loss for hard cases. 4) Prevailing contrastive learning: Including **self contrastive learning** (SelfCon) and **supervised contrastive learning** (SupCon) (Khosla et al., 2020). 5) **MixUp** (Sun et al., 2020): A typical interpolation-based data augmentation method.

6) HTC: We choose **HGCLR** (Wang et al., 2022d) and **HPT** (Wang et al., 2022c) because they are recent state-of-the-art methods in global HTC. Note that some baselines are supplementary rather than competitive counterparts because they can be jointly deployed with Text2Tree in practice (e.g., re-sampling).

**Implement Details.** For all the algorithms, we use *bert-base-uncased* for the English datasets and *bert-base-chinese* for the Chinese ones. We implement the experiment code with PyTorch on Python 3.8. All the experiments are run on NVIDIA RTX 3090. The optimizer is Adam for HG-CLR according to (Wang et al., 2022d) and is AdamW (Loshchilov and Hutter, 2018) for the others with the default configuration except for the learning rate. For hyperparameter tuning, we use gird search for each method with detailed hyperparameter spaces in Appendix C. Here we define the search space of HGCLR according to the recommended settings in (Wang et al., 2022d), and use default settings for HPT in (Wang et al., 2022c). We select hyperparameters with respect to Macro-F1 and Micro-F1 separately on the evaluation set. Due to the differences of the HTC label system (i.e., if predict "C01.784" MeSH label with HTC methods, the training framework should predict "C", "C01" and "C01.784" simultaneously), we consider extra higher level labels in HGCLR training loss and ignore them during metric calculation. The maximum token length is 512 for MIMIC-III and is 128 for the rest datasets. We uniformly adopt early stop with 10 epochs for fine-tuning based on the evaluation metric.

## 4.2 Main Results and Analyses

The main results are presented in Table 2. Our Text2Tree algorithm achieves the best Macro-F1 scores on four datasets, and stably ranks in the top 3 in both Micro-F1 and Macro-F1 across the five datasets, while the performances of the other methods are highly unstable. This demonstrates that incorporating label hierarchy to guide data reuse from other labels helps medical text representations essentially, especially for texts from samples of rare diseases (it helps Macro-F1).

We find that MixUp performs well on PubMed, which is probably attributed to the data scale, since random interpolation is more likely to produce more diverse data representations when the data source is sufficient. Hence, we can also see that it

performs badly on the small-sized Inpatient. Further, we see the best performance of SupCon on Inpatient, which may be due to the small data size with a large label space and detailed text semantics. The Inpatient dataset has the smallest scale but the most label amount and the longest average token length among the three multi-class datasets, and therefore is hard to directly learn ideal data representations. SupCon is able to precisely use label information to exploit fine-grained semantics in long texts, while Text2Tree is highly dependent on the quality of the learned label representations, which is inferior on the datasets in which all the labels are scarce (see Fig. 9, Appendix A).

As expected, we find that HGCLR and HPT usually perform inferiorly. As advanced hierarchical text classification (HTC) methods, they need to make prediction from all the levels in framework training, and thus would incur more redundancy to classify extra labels in flat text classification. Both of them perform relatively better on the multi-class datasets (i.e., the three real-world datasets) than on the multi-label ones (i.e., MIMIC-III and PubMed), since samples in multi-classification contain less higher level labels and thus the impact is weakened. Compared to HTC methods, our Text2Tree has more diverse contrastive signal because we treat any other sample as a potentially positive anchor. Besides, they takes more computational resource and training time, because HGCLR calculates classification signal on both the original samples and masked ones, doubling the training batch size, and HPT adopts the graph neural network for hierachy injection, which is computationally inefficient compared to attention-based HLR.

Although in this paper we experiment with Text2Tree on flat text classification in the medical domain, it can be easily extended to HTC settings or other general domains once the hierarchical label dependency is given (e.g., a code tree or a graph).

## 4.3 Ablation Study

To further validate the effectiveness of each key component in Text2Tree, we perform ablation study by separately removing one of 1) Similarity Surrogate Learning (**SSL**), 2) Dissimilarity Mixup Learning (**DML**), and 3) the Hierarchy-aware Label Representation Learning (**HLR**) module. Table 3 presents the ablation study results on two datasets (see Appendix E for the full results). Note that we detach the gradient in Eq. (10)

| Method | MIMIC-III | | PubMed | | Dermato. | | Gastro. | | Inpatient | |
|---|---|---|---|---|---|---|---|---|---|---|
| | Macro | Micro | Macro | Micro | Macro | Micro | Macro | Micro | Macro | Micro |
| Finetune | 42.22 | 51.65 | 57.17 | 65.30 | 53.93 | 56.31 | 47.64 | 50.52 | 71.22 | 71.12 |
| ROS (Menardi et al., 2014) | - | - | - | - | 53.40 | 53.76 | 44.52 | 45.08 | 72.44 | 72.28 |
| DBLoss (Huang et al., 2021) | 44.27 | **53.69** | 56.97 | 64.65 | - | - | - | - | - | - |
| FocalLoss (Lin et al., 2017) | 43.39 | 51.95 | 57.04 | 65.08 | 53.49 | 55.46 | 47.02 | 49.35 | 72.77 | 72.28 |
| SelfCon (Khosla et al., 2020) | 41.28 | 50.47 | 57.71 | 65.58 | 54.04 | 57.07 | 47.68 | 49.94 | 70.69 | 68.65 |
| SupCon (Khosla et al., 2020) | 42.21 | 52.02 | 57.53 | 65.43 | 54.69 | 56.83 | 47.66 | 50.90 | **73.44** | **73.43** |
| MixUp (Sun et al., 2020) | 42.11 | 51.72 | 58.29 | 65.82 | 54.58 | 56.16 | 48.40 | 49.64 | 70.98 | 70.79 |
| HGCLR (Wang et al., 2022d) | 41.59 | 51.53 | 55.77 | 64.50 | 53.34 | 56.50 | 48.53 | **51.70** | 71.07 | 71.29 |
| HPT (Wang et al., 2022c) | 41.32 | 51.11 | 57.64 | 65.73 | 53.82 | 56.02 | 47.58 | 49.51 | 73.08 | 72.96 |
| Text2Tree (ours) | **44.75** | 53.60 | **58.70** | **66.10** | **55.23** | **57.27** | **48.88** | 51.26 | 73.01 | 72.77 |

Table 2: Medical text classification results. The best results are marked in **bold** while the second best ones are underlined. "Marco" and "Micro" are for Macro-F1 and Micro-F1, respectively.

| Ablation | MIMIC-III | PubMed |
|---|---|---|
| Text2Tree | 44.75/53.60 | 58.70/66.10 |
| w/o SSL | 43.63/52.90 | 57.15/65.27 |
| w/o DML | 43.64/51.99 | 57.25/65.53 |
| w/o HLR | 43.07/52.47 | 57.90/65.57 |

Table 3: Ablation study results on the MIMIC-III and PubMed datasets. We report macro-F1/micro-F1 results when separately removing an individual component of Text2Tree. Ablation study results on all the five datasets are given in Appendix E.

(see Sec. 3.4), and thus HLR optimization is solely based on Eq. (8) in SSL. In the "w/o SSL" group, we retain the gradient in Eq. (10) to allow the HLR module to be optimizable. In the group "w/o HLR", there is no label representation, and thus we choose the combination of supervised contrastive learning (SupCon) and the ordinary mixup (MixUp) as substitution.

It can be seen that when keeping HLR, dropping either SSL or DML will harm the performances. It is hard to judge which of SSL and DML is more significant since they exhibit different levels of Macro-F1 or Micro-F1 decline across the five datasets.

When we deploy SupCon and MixUp jointly (the **w/o HLR** group), it is interesting to observe that this combination usually does not exceed the better single-method performance (the combination is worse than the better method across four datasets, see Appendix E and Table 2), which may be attributed to the incompatibility between MixUp and SupCon. The ordinary mixup randomly interpolates samples and may distort the fine-grained data representation space learned by SupCon, while our SSL and DML are respectively designed to prefer samples with more and less similar labels to alleviate this incompatibility, thus making HLR a necessary part of Text2Tree.

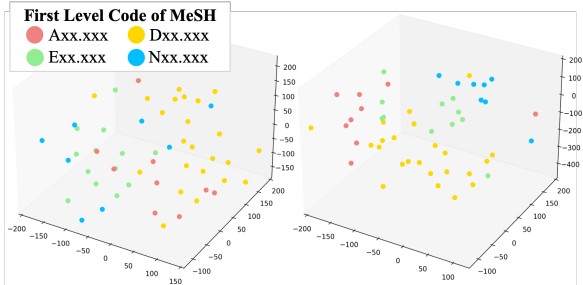

Figure 3: 3D t-SNE visualization of the label representations on the PubMed dataset, before (left) and after (right) the Text2Tree training. Labels with the same first-level codes are in the same color.

## 4.4 Visualization of Label Representations

To further illustrate the effectiveness of our HLR module, we visualize the label representations on the PubMed dataset (see Fig. 3). Here we choose four largest first level groups in 100 labels (MeSH codes starting with "A", "D", "E", "N") and simply assign the same color to labels of the same group. Starting with code (label) representations calculated from randomly initialized code (label) embeddings (the left figure), the HLR module indeed captures hierarchical bias with simple cascade attention modules in Sec. 3.1, and learns to cluster similar medical labels (the ones under the same first level). The SSL and DML processes are guided with reasonable label representations (the right figure) to reuse information from the other labels.

## 4.5 Effect on Rare Labels

To analyze the effect of Text2Tree on rare medical labels, we present Macro-F1 differences compared to the ordinary finetune on PubMed label groups when deploying different baselines (see Fig. 4). Overall, all the validated algorithms can boost performance on the rarest label group (labels 1~10), while DBLoss and FocalLoss are prone to

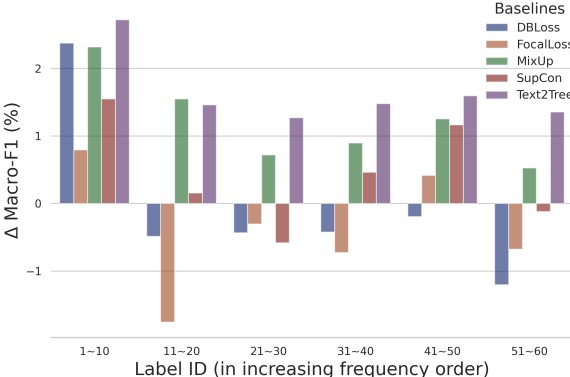

Figure 4: Macro-F1 changes on label groups (grouped by increasing-frequency ordered label IDs) on PubMed when adopting different algorithms to the ordinary fine-tune paradigm. Full results are given in Appendix E.

sacrificing generalization ability on more frequent labels (see their F1 changes on labels 11~100). SupCon is likely to attain conservative improvement on rare labels (labels 1~50) but fail to excel in common cases (labels 51~100). MixUp and `Text2Tree` show strong generalization on all the cases, while `Text2Tree` provides more stable and better F1 gains in most cases and is the best choice for the rarest labels.

## 5 Conclusions

In this paper, we proposed `Text2Tree`, a new framework-agnostic learning algorithm that aligns text representations to disease label hierarchy with two novel learning methods of contrastively reusing data, so as to resolve the data imbalance and scarcity issues in medical text classification. Compared with various state-of-the-art general imbalanced text classification methods, the superiority of `Text2Tree` was verified on 5 real-world datasets. We believe that `Text2Tree` will serve as a strong baseline in imbalanced medical text classification and could be extended to other domains when corresponding prior label dependency is provided.

## Limitations

While `Text2Tree` is able to exploit the label hierarchy to align medical text representations and achieve stable improvement among various classical and advanced imbalanced classification methods on the validated medical text classification datasets, one limitation is that `Text2Tree` explicitly needs prior label dependency to work, compared to the other general algorithms. Also, the

designed SSL and DML text representation learning methods require precise label representations. To provide such precision, we tailor the HLR module with lightweight cascade attention to introduce label hierarchy bias according to the code tree.

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

## A Detailed Dataset Information

Details of used datasets are shown in Table 4. We present label distributions of PubMed (Fig. 5), MIMIC-III (Fig. 6), Dermatology (Fig. 7), Gastroenterology (Fig. 8) and Inpatient (Fig. 9). Inpatient simulates extreme scarcity of medical data, where the most diseases have less than 30 records.

## B Data Clean Instructions

We introduce three real-world electronic medical record (EMR) datasets, including two outpatient record datasets and an inpatient record dataset. The outpatient records consist of 786,923 records from the Gastroenterology department and 635,302 records from the Dermatology department of a top hospital. The inpatient records are sourced from a healthcare institution, with a total of 9,032 records. To ensure the quality of the textual data, the following records are excluded:

- Duplicated records: Records with repetitive content are removed to avoid redundancy in the dataset.

- Trivial records: Records that contain simple follow-up visits or medication prescriptions, which provide no valuable information, such as "no change in medical history", "stable condition" or "continuing treatment" are eliminated.

- Label leakage: Records with descriptions that inadvertently reveal the diagnostic labels, such as "gastritis discovered during routine check-up", are excluded to prevent potential label leakage.

- Length-filtering: Records with a text length less than 15 characters are discarded as they lack enough information.

The main diagnosis in each medical record is extracted as the label and mapped to the ICD-10 coding system. Descriptions related to COVID-19, which were requested during the pandemic but lack relevant diagnostic information, are removed. Due to the evident long-tail distribution observed in the Gastroenterology and Dermatology records, only diseases with frequency greater than 50 in outpatient records and 20 in inpatient records are included in datasets.

## C Baseline Settings

### C.1 Implementation

For ROS we use the version implemented in *Imbalanced-learn* python package. For DBLoss, MixUp and HGCLR we reuse the implementation in the original paper (Huang et al., 2021; Sun et al., 2020; Wang et al., 2022d). For FocalLoss we extend the multi-class version in (Lin et al., 2017) to the multi-label version for MIMIC-III and PubMed. For SelfCon and SupCon, we reuse the implementation in (Khosla et al., 2020) and adapt the code from image classification task to text classification scenario.

### C.2 Hyperparameter Tuning

For DBLoss we use recommended setting in (Huang et al., 2021), we select hyperparameters of all baselines with grid search on the hyperparameter spaces provided in the following tables (Table 5~Table 9). We set learning rate space for all baselines to $\{5e-6, 1e-5, 3e-5, 5e-5\}$. We use batch size of 16 for MIMIC-III and 64 for the rest datasets.

## D Data and Gradient Flow in `Text2Tree`

Fig. 10 illustrates data flow and gradient flow of our used `Text2Tree`. Detaching gradient from DML is an empirical choice, for both DML and SSL can impact HLR, we concerned gradients from both will destabilize the learning process and select the best gradient strategy by macro-F1. We provide results on different graident policies in Table 10.

## E Additional Results

Table 11 reports results of ablation study on the five datasets. Fig. 11 presents Macro-F1 changes in all the label groups of PubMed for different baselines.

| Dataset | $N$ | $|Y|$ | Avg($l_i$) | Avg($|y_i|$) | # train | # dev | # test | task | C.S. | Lang. |
|---|---|---|---|---|---|---|---|---|---|---|
| MIMIC-III | 11368 | 33 | 450.61 | 4.01 | 8066 | 1573 | 1729 | multi-label | ICD10 | English |
| PubMed | 50000 | 100 | 122.88 | 8.52 | 32000 | 8000 | 10000 | multi-label | MeSH | English |
| Dermatology | 20522 | 59 | 44.97 | - | 13103 | 3256 | 4163 | multi-class | ICD10 | Chinese |
| Gastroenterology | 34952 | 35 | 58.31 | - | 22351 | 5574 | 7027 | multi-class | ICD10 | Chinese |
| Inpatient | 2603 | 98 | 69.22 | - | 1627 | 370 | 606 | multi-class | ICD10 | Chinese |

Table 4: Detailed dataset statistics. "C.S." means code system.

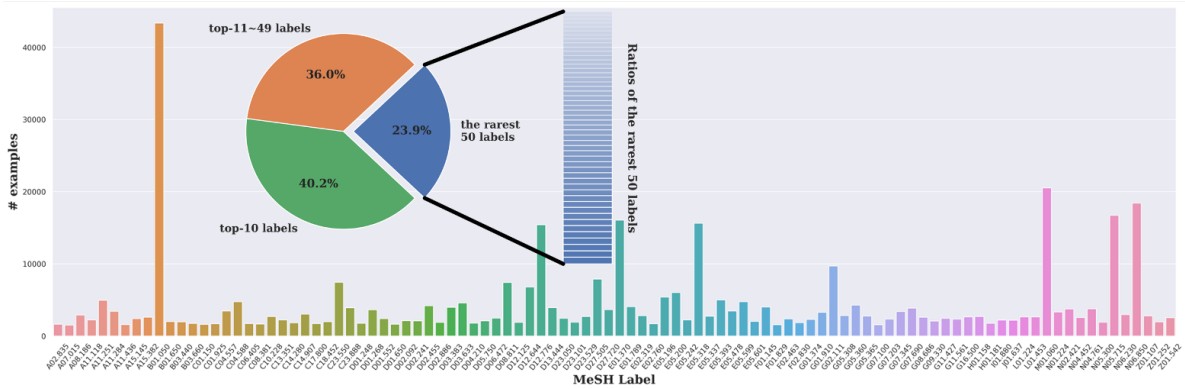

Figure 5: Data distribution of PubMed. The occurrence frequency of top-10 labels counts for 40.2 % in total top-100 labels, while the rarest 50 labels only account for 23.9 %.

| Parameter | Search space |
|---|---|
| $\alpha$ | $\{0.25, 0.5, 0.75\}$ |
| $\gamma$ | $\{0.5, 1.0, 2.0, 3.0\}$ |
| # iterations | $4 \times 3 \times 4 = 48$ |

Table 5: Hyperparameter space for FocalLoss.

| Parameter | Search space |
|---|---|
| temperature | $\{0.5, 1.0, 2.0, 3.0\}$ |
| $\lambda$ | $\{0.001, 0.005, 0.01, 0.05\}$ |
| # iterations | $4 \times 4 \times 4 = 64$ |

Table 6: Hyperparameter space for SelfCon and SupCon. $\lambda$ is a hyperparameter in controlling the loss weight (similar to Eq. (11)).

| Parameter | Search space |
|---|---|
| $\alpha$ | $\{0.1, 0.5, 1.0, 2.0, 4.0, 8.0\}$ |
| # iterations | $4 \times 6 = 24$ |

Table 7: Hyperparameter space for MixUp. $\alpha$ is the hyperparameter in Beta distribution.

| Parameter | Search space |
|---|---|
| $\gamma$ | $\{0.005, 0.01, 0.02, 0.05\}$ |
| $\lambda$ | $\{0.1, 0.3, 0.5, 1.0\}$ |
| temperature | 1.0 |
| # iterations | $4 \times 4 \times 4 = 64$ |

Table 8: Hyperparameter space for HGCLR. $\gamma$ and $\lambda$ are threshold and contrastive loss weight.

| Parameter | Search space |
|---|---|
| temperature | $\{0.5, 1.0, 2.0, 3.0\}$ |
| $\lambda$ | $\{0.001, 0.005, 0.01, 0.05\}$ |
| # iterations | $4 \times 4 \times 4 = 64$ |

Table 9: Hyperparameter space for Text2Tree.

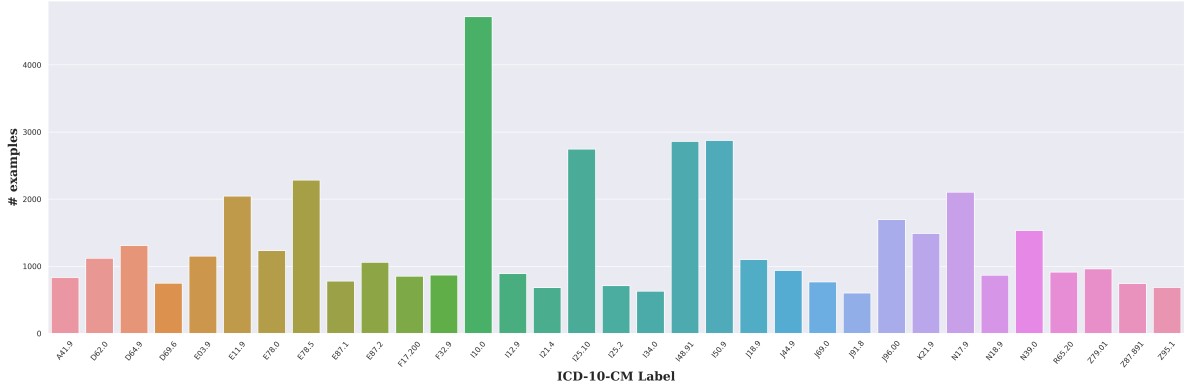

Figure 6: Data distribution of MIMIC-III. We only use top-50 labels and retain 33 disease labels in them.

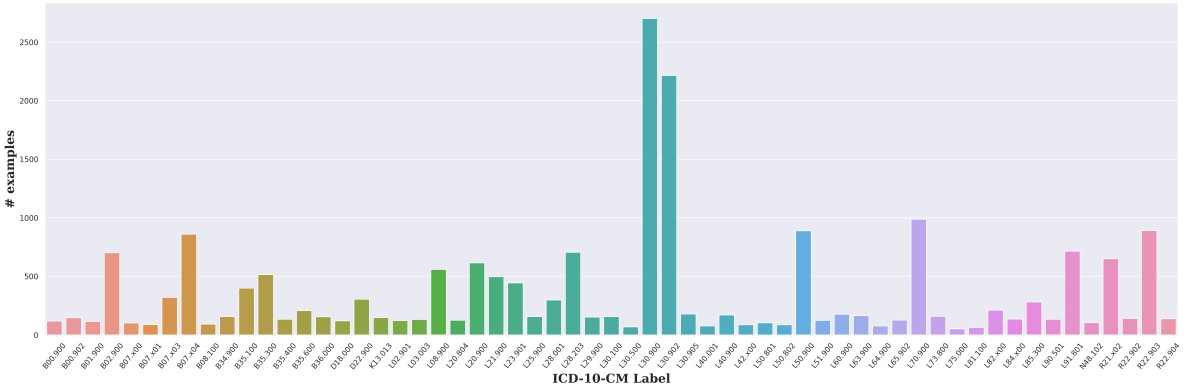

Figure 7: Data distribution of Dermatology. We cleaned 59 common diseases in dermatology department.

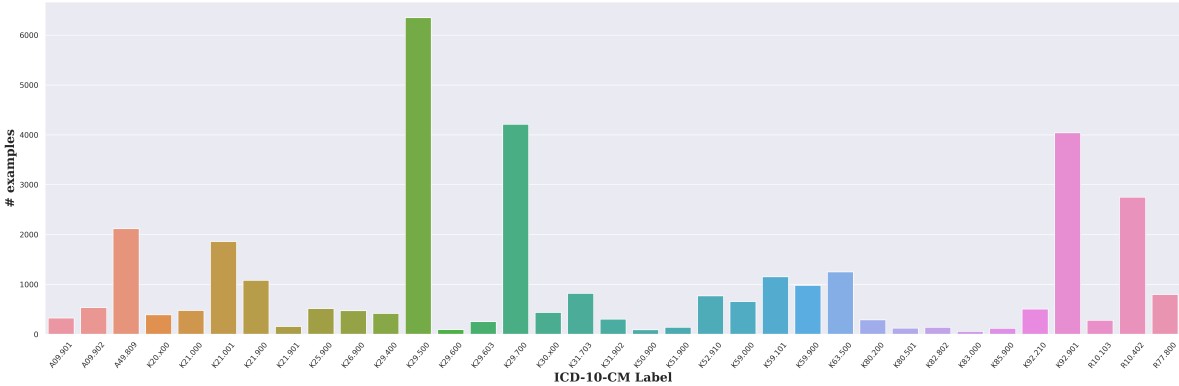

Figure 8: Data distribution of Gastroenterology. We cleaned 35 common diseases in gastroenterology department.

| Ablation | MIMIC-III | PubMed | Dermato. | Gastro. | Inpatient |
|---|---|---|---|---|---|
| det DML (baseline) | 44.75/53.60 | 58.70/66.10 | 55.23/57.27 | 48.88/51.26 | 73.01/72.77 |
| det SSL | 44.36/53.72 | 58.32/66.23 | 54.31/56.55 | 48.62/51.25 | 72.18/72.77 |
| no det | 44.13/53.38 | 57.87/66.13 | 54.12/57.84 | 47.92/51.00 | 72.83/72.28 |

Table 10: Results of different gradient policies on all the five datasets. det indicates gradient detach.

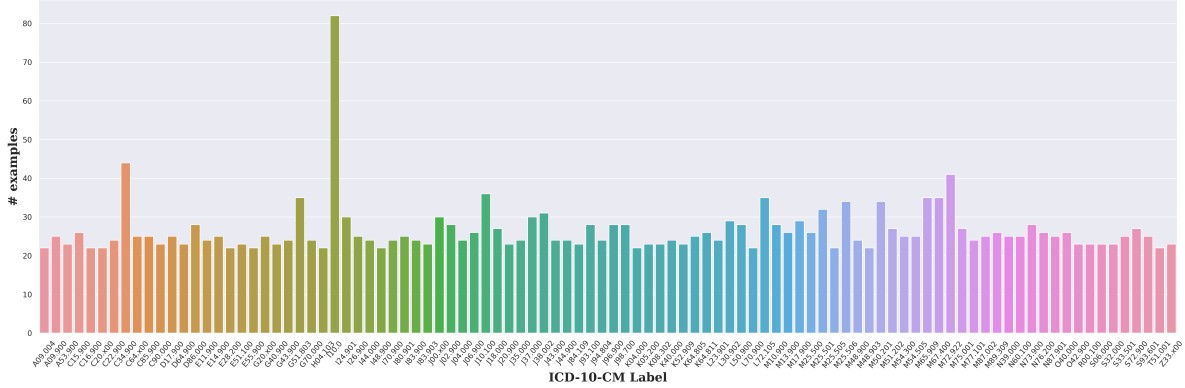

Figure 9: Data distribution of Inpatient. There are 98 diseases in the dataset of inpatient records, and the most diseases only have less than 30 records, leading to scarcity issue in each label.

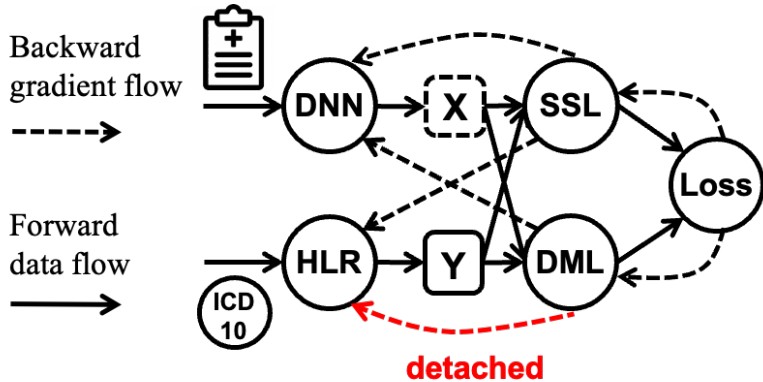

Figure 10: Forward data and backward gradient illustration of Text2Tree. We empirically detach the gradient from DML for stable learning.

| Ablation | MIMIC-III | PubMed | Dermato. | Gastro. | Inpatient |
|---|---|---|---|---|---|
| Text2Tree (baseline) | 44.75/53.60 | 58.70/66.10 | 55.23/57.27 | 48.88/51.26 | 73.01/72.77 |
| w/o SSL | 43.63/52.90 | 57.15/65.27 | 54.53/56.55 | 48.34/50.80 | 71.24/71.45 |
| w/o DML | 43.64/51.99 | 57.25/65.53 | 54.31/57.89 | 47.51/51.00 | 72.35/72.61 |
| w/o HLR | 43.07/52.47 | 57.90/65.57 | 53.95/55.22 | 47.19/50.29 | 71.50/71.78 |

Table 11: Ablation results on all the five datasets.

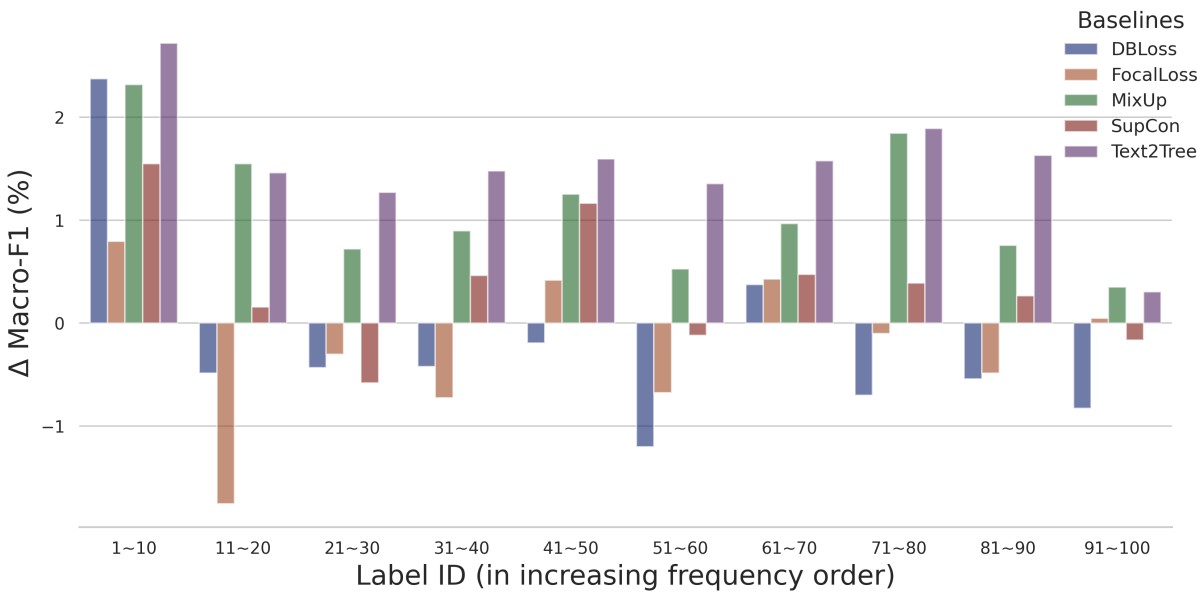

Figure 11: Full version of Macro-F1 changes on grouped labels (labels are ordered by their frequency and evaluation is performed on each label group, each group contains 10 labels) on PubMed when adopting different algorithms to the ordinary fine-tuning paradigm.