# OpenReview forum: "Text2Tree: Aligning Text Representation to the Label Tree Hierarchy for Imbalanced Medical Classification"
_EMNLP/2023/Conference — EMNLP 2023 Findings_

### Official Review · Reviewer_xmzY · 2023-07-21

**Typos Grammar Style And Presentation Improvements:** 1) Line 143 imblance -> imbalance
2) …
**Soundness:** 2

**Excitement:**

3: Ambivalent: It has merits (e.g., it reports state-of-the-art results, the idea is nice), but there are key weaknesses (e.g., it describes incremental work), and it can significantly benefit from another round of revision. However, I won't object to accepting it if my co-reviewers champion it.

**Paper Topic And Main Contributions:**

This paper proposes a Hierarchy Aligned Representation Learning (HARL) for medical text classification and attempts to address the imbalanced problem where some classes have insufficient training data. Besides an ordinary BERT and a linear classifier, HARL contains two more components: 1) Similarity Surrogate Learning adopts supervised contrastive learning by viewing all samples in a batch as weighted positive samples; 2) Dissimilarity Mixup Learning trains the model with weighted mixup samples. The authors use similarity scores of label representations as the weights for the two components, and the label representations are learned by a cascade attention module.

**Questions For The Authors:**

Question A: Why detach the gradient in Equation 10? It seems like the label representation module can be guided either by SSL or DML so why choose SSL instead of DML or use both?

Question B: What is the advantage of the cascade tree attention module? I see neither explanation nor experiments about why this is a better choice than a GAT or a Graphformer. In previous HTC works, graph encoders can have just one layer so they are more light-weighted.

Question C: What does j refer to in Equation 10? Does one sample mixup with every other sample in a batch or just a random one?

**Reasons To Accept:**

1) This paper first explores the medical text classification problem from a data imbalance perspective and comes up with a solution.
2) This paper proposes a general solution for contrastive learning with the imbalance problem, which can be adapted to any scenario with imbalanced training data.
3) Experiments show that every component of the proposed model is useful.

**Reasons To Reject:**

1) Section 3.2 is unfinished (line 365).
2) Strong baselines are missing. Although this paper is about medical text classification, this problem can be easily transformed into hierarchical text classification (HTC). Even though the authors select one of the HTC models, i.e. HGCLR, as a baseline, stronger HTC models fail to compare. Works like

   * Wang, Zihan, et al. "HPT: Hierarchy-aware Prompt Tuning for Hierarchical Text Classification." Proceedings of the 2022 Conference on Empirical Methods in Natural Language Processing. 2022.
   * Jiang, Ting, et al. "Exploiting Global and Local Hierarchies for Hierarchical Text Classification." Proceedings of the 2022 Conference on Empirical Methods in Natural Language Processing. 2022.

    achieve far better results than HGCLR so they should be compared. Among them, HPT (the first one) is adept at handling the imbalance and low resource situations, which may have solved the problem already.
3) The ablation results are weak. Although the authors demonstrate each component is useful, they fail to mention how each component is better than current solutions. E.g., how about using the normal contrastive learning loss instead of the proposed SSL or using an ordinary mixup strategy instead of DML? There is no experiment about how the proposed design is necessary instead of using existing works.

**Edit after rebuttal**

The authors explain the above three points in detail with lots of extra experiments. I will not change my score because it's unfair for papers that have done the experiments part well before the rebuttal.

**Reproducibility:**

4: Could mostly reproduce the results, but there may be some variation because of sample variance or minor variations in their interpretation of the protocol or method.

**Reviewer Confidence:**

4: Quite sure. I tried to check the important points carefully. It's unlikely, though conceivable, that I missed something that should affect my ratings.

---

> ### Author Rebuttal · Authors · 2023-08-29
>
> Thank you for finding our approach novel, useful and potentially generalized. We hope our replies address all of your concerns.
>
> > Q1: Section 3.2?
>
> A1: Thank you. We would like to state that derivation of Eq. (9) indicates our SSL loss leverages samples from other labels to contribute contrastive signal (since $L_{right}$ > 0 (Eq. (9)), and such extra signal is likely to relief data scarcity problem in rare disease labels by alleviating sparsity of positive samples.
> We will make it more clear in the final version. Thank you!
>
> > Q2: Weak baselines, strong HTC baselines missing?
>
> A2: Thank you for the question. Actually, we have referred the metioned models in the related work, see Line 221-222 (about HPT [1]) and Line 224-226 (about HBGL [2]) in Page 2.
>
> Besidse, we included HTC baseline HGCLR [3] for its using an ordinary finetune head and adopting conservative learning idea as HARL, while HPT and HBGL used prompt-tuning and a label tree MLM head separately. The clarification of baseline selection is stated at:
>
> - “We introduce various **framework-agnostic** algorithms for fair comparison”, in Line 431-433, Page 6.
> - “Group **ordinary finetune** is included”, in Line 434, Page 6. Baselines were all use ordinary finetune head.
>
> We highly agree with your reminding on insufficient advanced HTC baselines and are very willing to conduct additional experiments. Here are the results, “N” in “arch-specific” denotes HARL directly works on hidden states, though in the paper we used ordinary finetune on BERT, practically it can be adopted to any architectures e.g., prompt-tuning, semantic matching. We report average results on the same 5 random seeds in the paper.
>
> |       |  MIMIC-III  |   PubMed    |  Dermato.   |   Gastro.   |  Inpatient  |  arch-specific   |
> | :---- | :---------: | :---------: | :---------: | :---------: | :---------: | :--------------: |
> | HGCLR | 41.59/51.53 | 55.77/64.50 | 53.34/56.50 | 48.53/51.70 | 71.07/71.29 | ordinary finetune|
> | HBGL  | 42.65/52.37 | 57.38/65.03 | 53.61/55.90 | 47.36/50.02| 72.18/72.63  | label tree MLM |
> | HPT   | 41.32/51.11 | 57.64/65.73 | 53.82/56.02 | 47.58/49.51 | 73.08/72.96 | prompt-tuning |
> | HARL  | 44.75/53.60 | 58.70/66.10 | 55.23/57.27 | 48.88/51.26 | 73.01/72.77 |        N        |
>
> We will include these results in the final version. However, we would like to gently point out that we are working on medical text representation for the ICD coding task, specifically from the perspective of rare diseases, rather than HTC, and this topic have several key differences compared to the HTC field.
>
> - **Task Differnece**: ICD coding is typically a flat classification problem with all disease labels at the same level, whereas HTC techniques target multi-level label systems.
>
> - **Roadmap Difference**: HARL is an integrated data processing workflow (like well-known DIVIDEMIX [4]) that engages in flat classification instead of HTC.
>
> - **High Label Difficulty**: A typical ICD tree contains numerous parent nodes with sparse children, while in the commonly used HTC benchmark label trees, parents often possess many children.
>
> - **Key Contribution**: HARL firstly design a model-agnostic workflow with underlying ICD tree structure that focuses on imbalanced and scarce disease text and convey a processing spirit. Previous ICD coding works have never evaluated with HTC models \[6,7,8,9\], we are the first to include HTC baseline for completeness.
>
> - **Implementation Difference**: Actually, we use the cascade attention module for dynamical edge selection during training (we assumed edges among sibings are uncertain and selective, see “Q5&A5” for the design motivation and detailed technical differences), while HTC models typically use GNNs or Graphformer [5] that need pre-determined and static edges beforhand. Besides, label texts are used in HTC models but not used in our HARL.
>
> Here are baseline settings and analysis:
>
> - **Baseline Settings**: We follow graph strcture setting of \[1,2,3\] that assume only edges to parents are retained and used recommended model hyper-parameter spaces, other training settings are the same as HGCLR in our paper. We use `transformers==4.18.0` for HARL, HGCLR and HPT, and an old version `2.10.0` for HBGL as [HBGL GitHub](https://github.com/kongds/HBGL/blob/master). To faithfully follow the HTC label system, we reorganize ICD codes (i.e., turn “Virus Diseases(C01.925)” into “Diseases(C) -> Infections(C01) -> Virus Diseases(C01.925)”), HARL directly classifies leaf nodes without label texts and HTC models classify all (label texts accessible), early stop and final metrics are based on leaf nodes.
>
> - **Results**: In most cases, neither HPT nor HBGL can consistantly outputform HGCLR and HARL in ICD coding tasks, which might be caused by directly adopting HTC models to flat ICD coding task, and information from important sibling nodes is missing. Notably, HBGL and HPT achieve better performances on PubMed and Inpatient (than HGCLR), this may due to the great text abundance of medical abstracts and inpatient records, where label tree MLM and prompt-tuning are more suitable to match label words from input texts compared to ordinary finetuned methods.
>
> Thank you for your careful comments, we will clarify the detailed difference between medical text representation under ICD coding and HTC in the related works.
>
> > Q3: Weak ablation, fail to mention comparison with current solutions (normal contrastive loss & mixup) for SSL and DML separately?
>
> A3: Thank you for your careful concerns. We conducted ablations on:
>
> - “w/o SSL”, as stated in Sec. 4.3, Page 7-8, by removing SSL part from HARL.
> - “w/o DML”, as stated in Sec. 4.3, Page 7-8, by removing DML part from HARL.
> - “w/o HRL”, as stated in Line 545-548, Page 7, by constructing combination of normal supervised contrastive loss (SupCon) and mixup.
>
> All the ablation results demonstrate each component is well designed and improves performance. Following your suggestions, we add the further ablation groups:
>
> - “Compare SSL with normal contrastive loss” means we need a group like “- SSL + SupCon”.
> - “Compare DML with normal mixup” means we need a group like “- DML + mixup”.
>
> We highly agree with a further ablation and deeper analysis on single component of SLL and DML, and would like to conduct additional experiments to compare SSL with normal contrastive loss (see "- SSL + SupCon" below) and compare DML with normal mixup (- DML + mixup).
>
> |             |  MIMIC-III  |    PubMed   |  Dermato.  |   Gastro.   |  Inpatient  |
> | :---------- | :---------: | :---------: | :---------: | :---------: | :---------: |
> | HARL        | 44.75/53.60 | 58.70/66.10 | 55.23/57.27 | 48.88/51.26 | 73.01/72.77 |
> | -SSL+SupCon | 43.25/52.68 | 56.49/65.35 | 54.99/56.83 | 47.89/50.83 | 72.80/73.76 |
> | -DML+mixup  | 42.83/52.32 | 55.92/64.47 | 53.96/56.47 | 47.43/50.46 | 72.18/71.83 |
>
> It can be seen replacing DML with normal mixup will consistently hurt the performance, and in most cases it can be worse by replacing SSL with SupCon as well (except micro-F1 of Inpatient). Notably, substitution on DML will lead to more performance drop than the one on SSL, which is probably attributed to the direct impact of DML on text hidden states for final classification. Overall, replacing either SSL or DML with their normal solutions cannot benefit from hierarchy-aware label representation and hurts HARL performance.
>
> We will make addtional ablation analysis more clear and detailed in the final version.
>
> > Q4: Why detach in Eq. (10) (DML)? What about only detach SSL, or use gradients from both?
>
> A4: Thank you for the detailed question. HARL is trainable in all 3 cases (detach only DML or SSL, or use both gradients). Actually, this is an empirical choice, for both DML and SSL can impact HLR, we concerned gradients from both will destabilize the learning process and select the best gradient strategy by macro-F1. Here are results on 3 gradient policies.
>
> |                           |  MIMIC-III  |   PubMed   |  Dermato.  |   Gastro.   |  Inpatient  |
> | :------------------------ | :---------: | :---------: | :---------: | :---------: | :---------: |
> | det. DML(current) | 44.75/53.60 | 58.70/66.10 | 55.23/57.27 | 48.88/51.26 | 73.01/72.77 |
> | det. SSL                  | 44.36/53.72 | 58.32/66.23 | 54.31/56.55 | 48.62/51.25 | 72.18/72.77 |
> | no det.                   | 44.13/53.38 | 57.87/66.13 | 54.12/57.84 | 47.92/51.00 | 72.83/72.28 |
>
> We very appreciate for your mention on our undetailed gradient design, and would like to include and clear the empirical reason behind “detach SSL” in the final version. We have also prepared a data / gradient flow figure during HARL training process for friendly comprehension.
>
> > Q5: Advantages and necessity of using cascade attention module (CAM) in HLR compared to commonly used one-layer GAT or Graphformer in HTC works? Really light-weighted?
>
> A5: Thank you for your thoughtful comments. We have described the goal of CAM as:
>
> - “Selection of similar diseases from the same level to compensate for a scarce one”, as stated in Line 89-102, Page 2.
> - “Dynamical selection of sibling codes with parametric indicators during training”, as stated in Line 255-262, Page 4.
> - “Use information from the parent and selected siblings to represent a code”, as stated in Line 259-262, Page 4.
>
> We elaborate the necessity into three parts: design motivation, technical difference and advantages.
>
> - **Design Motivation**: There're two key designs of HLR.
>
>   1. Each ICD code compulsorily links its parent, this promises codes with common ancestors tend to form more similar representations because of shared ancestor information.
>
>   2. Each ICD code selectively links its siblings. It was controlled by parametric indicators in Eq. (2), with the motivation that codes in the same level are probably similar, e.g. fungal pneumonia is a scare disease but can share similar clinical representation with common virus pneumonia (e.g., COVID-19).
>
>   The motivation and design require to use a dynamical edge selection strategy for the label graph during training, while commonly used GNNs or Graphformer in HTC works \[1,2,3\] assume a pre-determined and static edges beforhand.
>
> - **Technical difference** between common graph encoder (e.g., GAT and Graphformer) and HLR are
>
>   1. Our HLR can dynamically select sibling codes with parametric indicators during training (as stated in Line 255-262, Page 4), while GNNs or Graphformer typically used pre-determined and static edges without connecting sibling codes [1,2,3]. Notably, though GAT can also dynamically computes the weights of each code, the edges are pre-determined and it has no chance to capture inter-sibling relationships.
>
>   2. Our HLR is able to select similar diseases from the same level to compensate for a rare one (as stated in Line 89-102, Page 2), while GNNs or Graphformer cannot help nodes fetch information from their sibling nodes.
>
> - **Advantages of CAM** compared to common graph encoder: Besides the ability of dynamical sibling selection (discussed above), CAM has better computation complexity (than Graphformer) and empirical training time.
>
>   1. Computation Complexity Analysis on Graphformer & CAM: Given an ICD tree with $N$ nodes, the complexity of a one-layer Graphformer is $O(N^2 + N)$, and a 3-layer CAM (assume $a,b,c$ nodes each layer) is $O(a^2+b^2+c^2)$, where $a+b+c \approx N$, the complexity difference can be more pronounced when $N$ is larger, for CAM maintains 3 small attention maps while Graphformer is a large one. Besides, Graphformer includes extra processing on spatial and edge encodings.
>
>   2. Empirical Training Backward Time Evaluation: To further empirically assess the computation efficiency of the CAM, we would like to include an additional experimental comparison of training backward time on five training sets, we use CAM, GAT, Graphformer to implement HLR separately (GAT and Graphformer follow graph structure as \[1,2,3\], substitution codes of GAT and Graphformer are from HPT GitHub repository), and average training backward time per sample is reported as follows.
>
>   3. Inference Time: After HARL training process, DNN models perform inference directly without HARL components and has no computation difference.
>
> | avg. train backward time per sample (s) | MIMIC-III (N=74,L=512) | PubMed (N=167,L=128) | Dermato. (N=107,L=128) | Gastro. (N=60,L=128) | Inpatient (N=206,L=128) |
> | :--------------------------------------------- | :---------------------------: | :-------------------------: | :---------------------------: | :-------------------------: | :----------------------------: |
> | HARL(3-layer CAM)                              |            0.0221            |           0.0056           |            0.0115            |           0.0094           |             0.0288             |
> | HARL(1-layer Graphformer)                      |            0.0382            |           0.0186           |            0.0322            |           0.0191           |             0.1153             |
> | HARL(1-layer GAT)                              |            0.0433            |           0.0192           |            0.0301            |           0.0182           |             0.1297             |
>
> In the table, the “N” denotes the code amount in the ICD tree, the “L” denotes maximum token length used for the dataset. From the dataset dimension, it is reasonable that MIMIC-III and Inpatient has the longer average training time than others for the long document and large label amount separately. From the method dimension, GAT and Graphformer need longer time to optimize, and this difference becomes even more pronounced when “N” is larger (e.g., on Inpatient, HARL with CAM is 4~5 times faster than the one with GAT or Graphformer).  Empirically, CAM is faster than GAT, this may benefit from the hardware acceleration for attention parallel calculation.
>
> We highly agree that the advantages and the necessity of using CAM implemented HLR should be emphatically discussed and demonstrated. We will include the detailed motivation, empirical evaluation to make the advantages more clear in the final version.
>
> > Q6: What $j$ means in Eq. (10)?
>
> A6: In Eq. (10), $j$ represents all other samples in a batch size. We follow the same formulaic symbols in the early works of mixup applications in NLP [10,11], and will add the denotation in the final version.
>
> #### References
>
> \[1\] Wang Z, et al. "HPT: Hierarchy-aware Prompt Tuning for Hierarchical Text Classification." EMNLP. 2022
>
> \[2\] Jiang T, et al. "Exploiting Global and Local Hierarchies for Hierarchical Text Classification." EMNLP. 2022
>
> \[3\] Wang Z, et al. "Incorporating Hierarchy into Text Encoder: a Contrastive Learning Approach for Hierarchical Text Classification." ACL. 2022
>
> \[4\] Li J, et al. "DivideMix: Learning with Noisy Labels as Semi-supervised Learning." ICLR. 2020
>
> \[5\] Ying C, et al. "Do transformers really perform badly for graph representation?." NeurIPS. 2021
>
> \[6\] Xie X, Xiong Y, et al. "EHR coding with multi-scale feature attention and structured knowledge graph propagation." CIKM. 2019
>
> \[7\] Chen Y, et al. "Automatic ICD code assignment utilizing textual descriptions and hierarchical structure of ICD code." BIBM. 2019
>
> \[8\] Cao P, et al. "HyperCore: Hyperbolic and co-graph representation for automatic ICD coding." ACL. 2020
>
> \[9\] Yang Z, et al. "Knowledge Injected Prompt Based Fine-tuning for Multi-label Few-shot ICD Coding." EMNLP. 2022
>
> \[10\] Chen J. "MixText: Linguistically-Informed Interpolation of Hidden Space for Semi-Supervised Text Classification." ACL. 2020
>
> \[11\] Sun L, et al. "Mixup-Transformer: Dynamic Data Augmentation for NLP Tasks." COLING. 2020

---

### Official Review · Reviewer_8duU · 2023-08-02

**Soundness:** 4

**Excitement:**

4: Strong: This paper deepens the understanding of some phenomenon or lowers the barriers to an existing research direction.

**Paper Topic And Main Contributions:**

This paper addresses the problem of medical text classifications from the perspective of sample imbalance and scarcity. The main contribution of this paper is that the proposed HARL algorithm realizes the hierarchical alignment of textual and labeled representations and establishes a benchmark for the classification of unbalanced medical texts. In this paper, the effectiveness of the method is demonstrated through adequate medical text classification experiments and ablation experiments.

**Questions For The Authors:**

Question A: What is the performance of the HARL algorithm proposed in this paper? How does it compare with the baseline approaches?
Question B: Does the baseline method use any other metrics as evaluation criteria?

**Reasons To Accept:**

1. Sample imbalance and scarcity are common problems in medical text classification. The task is worth studying.
2. The idea of incorporating code-embedded trees to guide the expression of medical texts is reasonable and could be applied to some relative tasks.
3. The experiments demonstrate the effectiveness of the proposed approach on 5 real-world datasets.


**Reasons To Reject:**

1. The description of the method is not comprehensive or detailed. For example, use an overall framework diagram to clarify the interactions between the three modules（HLR, SSL&DML）.
2. In the experimental part of this paper, the DBLoss method performs close to the method proposed in this paper on the MIMIC-III dataset and lacks analyses.
3. The writing of this paper should be improved.

**Reproducibility:**

3: Could reproduce the results with some difficulty. The settings of parameters are underspecified or subjectively determined; the training/evaluation data are not widely available.

**Reviewer Confidence:**

3: Pretty sure, but there's a chance I missed something. Although I have a good feel for this area in general, I did not carefully check the paper's details, e.g., the math, experimental design, or novelty.

---

> ### Author Rebuttal · Authors · 2023-08-29
>
> Thank you for your postive comments and finding our topic important, proposed approach reasonable and experiment adequate. We hope our responses can clarify all your concerns.
>
> > Q1: The description of the method is not detailed.
>
> A1: Thank you. HARL consists of HLR, SSL and DML, in which SSL and DML utilize label representations from HLR. We describe HLR process and relationship of three components separately.
>
> 1. **HLR Process**: In this work, HLR is implemented with cascade attention module (CAM). We have prepared an additional figure for illustration of processing a 2-level label tree in the final version. Here we elaborate the HLR process literally.
>
>    We denote label embeddings as the symbol $e$ and representations as $h$. Each label's $h$ is acquired from attention calculation with its $e$ as query, its parent’s $h$, selected siblings’ $e$ and its $e$ as keys and values. All $e$ are randomly initialized. The attention process is performed layer by layer, and the final outputs of the CAM are leaf node labels' $h$, which are further used to calculate pair-wise label similarity in SSL and DML. Each attention layer processes only the labels of adjacent levels. There're two key designs:
>
>     - Each label compulsorily links its parent, this promises labels with common ancestors tend to form more similar $h$ because of shared ancestor information.
>     - Each label selectively links its siblings, which is controlled by parametric indicators in Eq. (2), with the motivation that labels of the same level are probably similar, e.g., fungal pneumonia is a scare disease but can share similar clinical representation with common virus pneumonia (e.g., COVID-19).
>
>     **HLR Contributions**: HLR takes the advantage of this design to help the rare disease code fetch important information from its sibling codes and parent code, while traditionally used graph encoders for the label tree only considered information from parent label. We firstly designed and applied HLR that can dynamically select sibling codes in the label tree.
>
> 2. **Component Relationship**: We have prepared a data / gradient flow figure during HARL training process to illustrate the relationship of three components in the final version. Here, we provide a detailed explanation of this relationship.
>
>    In forward process, HLR will embed input ICD codes with ICD label tree and output code embeddings to SSL and DML. SSL and DML perform code similarity based learning and dissimilarity based learning on medical text hidden states (encoded by any DNN text encoder), and jointly contribute the final loss. In backward process, we empirically detach the gradient from DML to HLR, considering gradient from both SSL and DML may unstabilize the learning process. For model inference, DNN directly performs ICD coding prediction task as usual.
>
> We highly agree that the HARL framework presentation should be more detailed. We will diligently polish and included prepared figures and elaboration for friendly comprehension.
>
> > Q2: Analysis of DBLoss based performances on the MIMIC-III dataset.
>
> A2: Thank you for your thoughtful suggestion. DBLoss is firstly proposed to solve long-tailed multi-label classification by reducing redundant information of label co-occurrence [1]. The probable reason it performs well is that MIMIC-III includes detailed inpatient medical records annotated with many coexisting diseases, where scenario label co-occurrence is frequent. However, when label co-occurrence is irregular, or each sample has only one label, DBLoss cannot take advantage, while HARL is fueled by label representations that can be applied to both multi-label or multi-class data if label information is accessible. For instance, on PubMed dataset (in fine-grained label version), label co-occurrence is spare and complicated, where DBLoss underperforms other baselines. Besides, DBLoss is unable to be deployed in multi-class datasets.
>
> We would like to further discuss and clearify performance of DBLoss in the final version, thank you!
>
> > Q3: The writing should be improved.
>
> Thank you. We are currently collaborating with a native-speaking professor on certain projects, and he has kindly agreed to assist us in refining this paper. We believe its final version will be more clear and easily understood.
>
> > Q4: HARL performance and comparison methods with baselines?
>
> A4: We will present HARL performance and baseline comparison methods separately here.
>
> 1. **Elaboration of HARL Performance**: We use macro-F1 & micro-F1 on five real-world medical datasets on ICD coding task to to evaluate HARL and baselines, HARL achieves best macro-F1 on 4 datasets and best micro-F1 on 3 datasets, and stably ranks top 2 in across all datasets (in Table 2, page 7). Besides, all macro-F1 gains of HARL on PubMed scarce label groups (label 1~90) stably exceed 1\% while other ordinary methods are unstable (in figure 10, page 14).
>
> 2. **Baseline Comparison**: We elaborate our comparison methods into baseline selection and settings.
>
>    - Baseline Selection: HARL is a model-agnostic workflow, thus we select 6 model-agnostic algorithms from four different roadmaps. For completeness, we firstly include an extra baseline from recent research of hierarchical text classification (HTC) [2], which is overlooked in previous ICD coding works.
>    - Baseline Settings: All baselines and HARL use the same BERT base model and ordinary finetune head for fair comparison, with each baseline using a recommended hyper-parameter space.
>
> > Q5: Other evaluation criteria for baseline methods?
>
> A5: In order to visualize the performance promotion on rare labels, we reported and analyzed baseline macro-F1 gains on PubMed label groups (grouped in frequency order) as macro score is more important for health care fairness and rare disease detection. We will add other clinical metrics (e.g., brier score) in the final version.
>
> #### References
>
> \[1\] Huang, Yi, et al. "Balancing methods for multi-label text classification with long-tailed class distribution." EMNLP. 2021
>
> \[2\] Wang, Zihan, et al. "Incorporating Hierarchy into Text Encoder: a Contrastive Learning Approach for Hierarchical Text Classification." ACL. 2022

---

### Official Review · Reviewer_LHTo · 2023-08-05

**Soundness:** 3

**Excitement:**

3: Ambivalent: It has merits (e.g., it reports state-of-the-art results, the idea is nice), but there are key weaknesses (e.g., it describes incremental work), and it can significantly benefit from another round of revision. However, I won't object to accepting it if my co-reviewers champion it.

**Missing References:**

[3] GRAM:  Graph-based Attention Model for Healthcare Representation Learning, Edward Choi et al., KDD 2017

**Paper Topic And Main Contributions:**

The authors address the issues of data imbalance and scarcity in biomedical classification. To this end, they propose a novel, framework-agnostic algorithm called Hierarchy Aligned Representation Learning (HARL). HARL captures the hierarchical structure of labels with a cascaded tree attention module, further applying it to contrastive learning. In particular, the authors propose a novel training objective called "similarity surrogate learning," which allows the use of all samples in a batch as negative examples. Additionally, they introduce a new mixup strategy to reuse sample information from less similar labels. On five biomedical text classification datasets (MIMIC-III, PubMed, Dermatology, Gastroenterology, and Inpatient), HARL exhibits similar or slightly better performance compared to the baseline. The authors evaluate HARL and baselines specifically for rare labels, demonstrating that HARL outperforms baselines in these settings.

**Questions For The Authors:**

Questions related to HLR
+ Is the HLR applied to each level just once? Could you explain the motivation behind this particular design choice?"
+ GRAM [3] also used graph neural networks to represent medical ontologies.How does your proposed approach offer key advantages compared to GRAM?
+ HLR incorporates information from both parent and sibling nodes, but what would occur if it were to aggregate information from a single direction? That is, only from either the parent or sibling nodes.

**Reasons To Accept:**

+ This work addresses a very practical problem and the proposed approaches are well designed.
+ Section 4.1 and Appendices B and C provide a thorough description of the experimental setting and implementation details, allowing the reimplementation of results.
+ The ablation study demonstrates that each design component proposed by the authors contributes to performance improvement. They also highlight that hierarchy-aware label representation learning (HLR) and other contrastive learning methods are incompatible. Moreover, the proposed approach works better on the rare labels compared to baselines.
+ The proposed approach demonstrates better performance on the classification of rare labels compared to baselines.


**Reasons To Reject:**

+ The contents in L355-365, which are crucial for understanding the proposed SSL, are incomplete. Specifically, (1) the description of Equation (9) is insufficient and (2) the contribution of SSL is not articulated.
+ The proposed HLR differs significantly from conventional graph neural networks such as GCN [1] and GAT [2], but the advantages of HLR are not well-explained in the paper. (Please refer to the Questions section for further discussion.)
+ The authors compute the similarity between two labels based on the graphical structure. However, since each label also has its own name and description, it would be possible to compute similarity based on natural language representations. To convincingly demonstrate the benefits of hierarchical label representation, the authors should conduct similar experiments using sentence similarity.
+ The performance gain from the proposed approach is small, and this is also true for rare labels.

[1] T. Kipf and M. Welling, Semi-Supervised Classification with Graph Convolutional Networks, ICLR 2017

[2] P. Velickovic et al., Graph Attention Networks, ICLR 2018

**Reproducibility:**

4: Could mostly reproduce the results, but there may be some variation because of sample variance or minor variations in their interpretation of the protocol or method.

**Reviewer Confidence:**

3: Pretty sure, but there's a chance I missed something. Although I have a good feel for this area in general, I did not carefully check the paper's details, e.g., the math, experimental design, or novelty.

**Typos Grammar Style And Presentation Improvements:**

L365: incomplete sentence
Caption of Table 2: “Marco” -> “macro”

---

> ### Author Rebuttal · Authors · 2023-08-29
>
> We would like to sincerely thank you for finding our task very practical, our approach novel and well designed. We hope the point-by-point responses can clarify all your concerns.
>
> > Q1: Contents in L355-365 are incomplete. The description of Equation (9) is insufficient and the contribution of SSL is not articulated.
>
> A1: Thanks for your thoughtful and careful comments. We would like to state that our SSL loss leverages samples from other labels to contribute contrastive signal (since $L_{right}$ > 0 (Eq. (9)), and such extra signal is likely to relief data scarcity problem in rare disease labels by alleviating sparsity of positive samples. Following your suggestion, we would like to explain Equation (9) and summarize the contribution of SSL by comparing to normal supervised contrastive loss (SupCon) as follows.
>
> - **Equation (9) Explanation**: Derivation of Eq. (9) indicates a progressive relationship that SSL is a generalized form of commonly used SupCon, i.e., when we treat any other samples in a batch size (or in a dataset) as potentially positive anchors (soft anchor), SupCon is generalized into SSL. In this paper, the weight between a sample and its soft anchor is their pair-wise label similarity (see Eq. (7)), which is normalized to 0~1. Hence, when a soft anchor has the same label with one sample, the weight is 1, and all soft anchors of the same label constitute the left loss ($L_{left}$, in Eq. (9)), which is equal to SupCon. The right loss ($L_{right}$, in Eq. (9)) represents contrast signal from samples with different labels. SSL can be very flexible with user-defined sample similarity function.
>
> - **Contributions of SSL**: In this work we define sample similarity as pair-wise label similarity. (1) If a sample is from common classes, a large amount of contrast signal comes from anchors with the same label, where SSL is approximately equivalent to SupCon; (2) When a sample in low resource like rare diseases, our SSL is helpful to alleviate sparsity of contrast signal by treating samples with similar labels as a soft (positive) anchor, while normal SupCon cannot provide supervision signal for optimization in this case. (3) Moreover, when hareware is limited (e.g., online learning in mobile phone), the training batch size is small, where SupCon may be ineffective but SSL promise a better upper bound of optimization space. By using our designed SSL rather than SupCon, contrastive signal becomes tractable for hard samples from rare labels.
>
> We will refine the contribution of SSL and explain Eq. (9) more clear in the final version. Thank you!
>
> > Q2: (1) The proposed HLR differs significantly from conventional graph neural networks such as GCN [1] and GAT [2], but the advantages of HLR are not well-explained in the paper. (Please refer to the Questions section for further discussion.) (2) Key advantages compared to GRAM [3]?
>
> A2: Thank you for your insightful concerns and identifying the distinctions of our HLR (implemented with cascade attention module, CAM) from GNNs.
>
> **As for question (1)**, there're two key designs of HLR:
>
> - Each ICD code compulsorily links its parent, this promises codes with common ancestors tend to form more similar representations because of shared ancestor information.
>
> - Each ICD code selectively links its siblings. It was controlled by parametric indicators in Eq. (2), with the motivation that codes in the same level are probably similar, e.g. fungal pneumonia is a scare disease but can share similar clinical representation with common virus pneumonia (e.g., COVID-19).
>
> The motivation and design require to use a dynamical edge selection strategy for the label graph during training, while ordinary GNNs assume a pre-determined static graph structure.
>
> The differences between GNNs and HLR are:
>
> 1. Our HLR can dynamically select sibling codes with parametric indicators during training (as stated in Line 255-262, Page 4), while GNNs applied in label tree typically used pre-determined and static edges without connecting sibling codes [5]. Notably, though GAT can also dynamically computes the weights of each code, the edges are pre-determined and it has no chance to capture inter-sibling relationships.
>
> 2. Our HLR selects similar diseases from the same level to compensate for a scarce one (as stated in Line 89-102, Page 2), while GNNs applied in other medical concept tree (e.g. GRAM [3]) cannot help nodes fetch important information from their sibling nodes.
>
> 3. Our HLR empirically outperforms commonly used graph encoders in computation efficiency. To assess the training speed of HLR (CAM implemented), we compared the training times of CAM-based, GAT-based, and Graphformer-based [4] HLR on the five datasets. Following the label tree structure of [5], on a GTX 3090Ti, the average training backward time per sample is shown in the table below. It can be clearly seen that our proposed CAM is empirically faster during training, and the superiority becomes even more pronounced when the code amount in ICD tree is larger (e.g., the Inpatient dataset has more than 200 codes in the tree and achieve the most substantial acceleration).
>
> | avg. train backward time per sample (s) | MIMIC-III | PubMed | Dermato. | Gastro. | Inpatient |
> | :------------------------------------------- | :-------: | :----: | :------: | :-----: | :-------: |
> | HARL(3-layer CAM, ours)                      |  0.0221  | 0.0056 |  0.0115  | 0.0094 |  0.0288  |
> | HARL(1-layer Graphformer)                    |  0.0382  | 0.0186 |  0.0322  | 0.0191 |  0.1153  |
> | HARL(1-layer GAT)                            |  0.0433  | 0.0192 |  0.0301  | 0.0182 |  0.1297  |
>
> We will highlight the differences between HLR and GNNs and include the empirical comparison in the final version. Thank you again for your constructive suggestions!
>
>
> **As for question (2)**, the graph-based attention in GRAM was applied in sequential diagnosis prediction where input features are structured patient visit (standard medical concepts in features are available), while HARL was used in ICD coding task with unstructured medical texts, which is vastly different in terms of application. If the graph-based attention could be used in ICD coding task, their key differences and our advantages would be:
>
> - **Key Difference**: “HARL incorperates label embedding tree to guide medical text representations”, as stated in Line 106-111, Page 2. HARL organizes knowledge in ICD label level to guide text feature learning, while GRAM operates at the feature level to augment and interact features with medical ontologies.
>
> - **Advantage**: As discussed in question (1), our HLR component uses a dynamical edge selection strategy, which may help to pass and learn more information. In contrast, GRAM was implemented in a GNN manner with static graph structure and did not contain edges among siblings. Such static graph structure likely misses additional information, especially when the amount of sibling nodes is large.
>
> > Q3: Additional experiments with label description (sentence) similarity?
>
> A3: Thank you for your suggestion. We initialized all label embeddings (in HLR) randomly to demonstrate the effectiveness obtained by only utilizing label tree structure. To further demonstrate the superiority of our HLR, we conduct additional experiment results on HLR using: (a) label hierarchy only (randomly init., in the paper) (b) label description only (fixed embeddings).
>
> |                        |  MIMIC-III  |   PubMed    |  Dermato.   |   Gastro.   |  Inpatient  |
> | :--------------------- | :---------: | :---------: | :---------: | :---------: | :---------: |
> | (a)rand. init.(out HLR)| 44.75/53.60 | 58.70/66.10 | 55.23/57.27 | 48.88/51.26 | 73.01/72.77 |
> | (b)text semantics only | 42.89/51.76 | 58.45/66.24 | 55.08/56.61 | 47.56/50.79 | 71.37/71.62 |
>
> It can be seen group (b) often underperforms our HLR. This outcome could potentially be attributed to the precision of label descriptions that some meaningful texts are sensible for humans but tough for a language model, whose text representation ability relies on the pre-training corpus. Furthermore, the embeddigs of group (b) remain fixed. Compared to the uncontrollable label texts, the label hierarchy is explicit enough to lead label embedding learning where HARL benefits significantly. We will include and discuss on these results in the final version.
>
> > Q4: The performance gain from the proposed approach is small？
>
> A4: Thank you for your careful concerns.
>
> 1. **Main Focus & Performance**: We would like to gently point out that this work focuses on coping with scarce diseases and imbalanced classification in the ICD coding tasks with model-agnostic manner. We have actually and stably improved the performance on scarce diseases. See figure 10, page 14, all macro-F1 gains of HARL on PubMed scarce label groups (label 1~90) stably exceed 1\% while compared methods are unstable. In Table 2, page 8, HARL stably outperforms across five real-world datasets while other baselines are data-biased solutions. Notably, since HARL focus on scarce diseases for health care fairness, the stable performances on metric macro could be more important.
>
> 2. **Comparison on Previous Works**: In fact, this task is of great significance and is extensively researched in previous works, because every bit of progress will further advance medical equity, genuinely enabling patients to break free from suffering [6,7,8]. The previous works were solely evaluated on MIMIC dataset, while we conducted extensive evaluations on multiple real-world datasets from hospitals to demonstrate a more stable and conving improvement, which is critical and beneficial to clinical practice.
>
> 3. **Significant Potential**: Since HARL is a model-agnostic workflow that can be unconditionally applied and improve any neural architectures for medical text. For fair comparison, we employ a simple base model (normally finetuned on a BERT) to emphasize the contributions stemming from the approaches we have devised. Thus, HARL holds considerable significance, as it can be employed in conjunction with other approaches or models to acquire further performance improvement and universal benefits.
>
> > Q5: How HLR is applied to each label level and the design motivation?
>
> A5: HLR learns label embeddings with cascade attention module (CAM), we addtionally prepared a figure for illustration of processing a 2-level label tree, which will be added in the final version. Here we elaborate the HLR process literally.
>
> We denote label embeddings as symbol $e$ and representations as $h$. Each label's $h$ is acquired from attention calculation with its $e$ as query, its parent’s $h$, selected siblings’ $e$ and its $e$ as keys and values. All $e$ are randomly initialized. The attention is performed layer by layer and the final output of CAM are leaf node labels' $h$, which are further used to calculate pair-wise label similarity in SSL and DML. Each attention layer only processes labels of adjacent levels. There're two key designs:
>
> - Each label compulsorily links its parent, this promises labels with common ancestors tend to form more similar $h$ because of shared ancestor information.
> - Each label selectively links its siblings, which is controlled by parametric indicators in Eq. (2), with the motivation that labels of the same level are probably similar, e.g. fungal pneumonia is a scare disease but can share similar clinical representation with common virus pneumonia (e.g., COVID-19).
>
> We would like to include detailed illustration and elaboration for friendly comprehension in the final version.
>
> > Q6: What if aggregate information from either parent or sibling node rather than both?
>
> A6: We are glad to hear your keen consideration. We discussed design motivation of dynamical sibling selection in the responses for “Q2 (question (1))” and “Q5”. Although solely considering parent or sibling may miss extra information, we would like to add experiments on solely aggregating from parent or siblings.
>
> |               |  MIMIC-III  |   PubMed   |  Dermato.  |   Gastro.   |  Inpatient  |
> | :------------ | :---------: | :---------: | :---------: | :---------: | :---------: |
> | both          | 44.75/53.60 | 58.70/66.10 | 55.23/57.27 | 48.88/51.26 | 73.01/72.77 |
> | parent only   | 44.38/53.12 | 57.89/65.84 | 55.36/57.12 | 48.23/50.87 | 72.98/73.06 |
> | siblings only | 43.25/52.09 | 57.41/65.13 | 54.60/56.87 | 48.46/50.73 | 71.95/71.38 |
>
> It can be seen only including siblings is often less effective than other case. This observation aligns with our motivation, as not all siblings contribute beneficial information, and that is the reason we need to select them in a data-driven manner. Conversely, only including parent ensures a better performance than only relying on siblings, as it promises an errorless information source.
>
> #### References
>
> \[1\] T. Kipf and M. Welling, "Semi-Supervised Classification with Graph Convolutional Networks", ICLR. 2017
>
> \[2\] P. Velickovic et al., "Graph Attention Networks", ICLR. 2018
>
> \[3\] Edward Choi et al. "GRAM: Graph-based Attention Model for Healthcare Representation Learning", KDD. 2017
>
> \[4\] Ying C, et al. "Do transformers really perform badly for graph representation?." NeurIPS. 2021
>
> \[5\] Wang Zi, et al. "Incorporating hierarchy into text encoder: a contrastive learning approach for hierarchical text classification." ACL. 2022
>
> \[6\] Xie X, Xiong Y, et al. "EHR coding with multi-scale feature attention and structured knowledge graph propagation." CIKM. 2019
>
> \[7\] Cao Pengfei, et al. "HyperCore: Hyperbolic and co-graph representation for automatic ICD coding." ACL. 2020
>
> \[8\] Yang Z, et al. "Knowledge Injected Prompt Based Fine-tuning for Multi-label Few-shot ICD Coding." EMNLP. 2022

---

### Meta-Review · Area_Chair_MZLr · 2023-09-18

**Recommendation:** 2

**Metareview:**

This paper proposes HARL, a method for performing the automatic coding task (i.e. predicting diagnosis codes based on clinical text). HARL uses attention on diagnosis label taxonomy to learn useful features for rare/imbalance diagnosis codes, and empirically showed favorable performance. All reviewers agreed on the value of the task itself, and the performance is promising, there were several strong concerns such as: 1) HARL's performance on predicting rare diagnosis codes (which is the core motivation of this paper) are comparable to baselines; 2) HARL makes an exaggerated claim of novelty regarding the use of diagnosis code hierarchy, which has been explored multiple times before in the healthcare domain 3) HARL fails to accurately position its core claim between medical text classification and general imbalanced classification algorithm.

---

### Decision · Program_Chairs · 2023-10-07

**Decision:**

Accept-Findings

**Comment:**

This paper proposes HARL, a method for performing the automatic coding task (i.e. predicting diagnosis codes based on clinical text). HARL uses attention on diagnosis label taxonomy to learn useful features for rare/imbalance diagnosis codes, and empirically showed favorable performance. All reviewers agreed on the value of the task itself, and the performance is promising, there were several strong concerns such as: 1) HARL's performance on predicting rare diagnosis codes (which is the core motivation of this paper) are comparable to baselines; 2) HARL makes an exaggerated claim of novelty regarding the use of diagnosis code hierarchy, which has been explored multiple times before in the healthcare domain 3) HARL fails to accurately position its core claim between medical text classification and general imbalanced classification algorithm.